# Assessing racial bias in type 2 diabetes risk prediction algorithms

**Héléne T. Cronjé**, **Alexandros Katsiferis**, **Leonie K. Elsenburg**, **Thea O. Andersen**, **Naja H. Rod**, **Tri-Long Nguyen**, **Tibor V. Varga** *

Section of Epidemiology, Department of Public Health, University of Copenhagen, Copenhagen, Denmark

☙ These authors contributed equally to this work.
* tibor.varga@sund.ku.dk

**Data Availability Statement:** NHANES data is publicly available and can be accessed via https://www.cdc.gov/nchs/nhanes/index.htm.

## Abstract

Risk prediction models for type 2 diabetes can be useful for the early detection of individuals at high risk. However, models may also bias clinical decision-making processes, for instance by differential risk miscalibration across racial groups. We investigated whether the Prediabetes Risk Test (PRT) issued by the National Diabetes Prevention Program, and two prognostic models, the Framingham Offspring Risk Score, and the ARIC Model, demonstrate racial bias between non-Hispanic Whites and non-Hispanic Blacks. We used National Health and Nutrition Examination Survey (NHANES) data, sampled in six independent two-year batches between 1999 and 2010. A total of 9,987 adults without a prior diagnosis of diabetes and with fasting blood samples available were included. We calculated race- and year-specific average predicted risks of type 2 diabetes according to the risk models. We compared the predicted risks with observed ones extracted from the US Diabetes Surveillance System across racial groups (summary calibration). All investigated models were found to be miscalibrated with regard to race, consistently across the survey years. The Framingham Offspring Risk Score overestimated type 2 diabetes risk for non-Hispanic Whites and underestimated risk for non-Hispanic Blacks. The PRT and the ARIC models overestimated risk for both races, but more so for non-Hispanic Whites. These landmark models overestimated the risk of type 2 diabetes for non-Hispanic Whites more severely than for non-Hispanic Blacks. This may result in a larger proportion of non-Hispanic Whites being prioritized for preventive interventions, but it also increases the risk of overdiagnosis and overtreatment in this group. On the other hand, a larger proportion of non-Hispanic Blacks may be potentially underprioritized and undertreated.

## Introduction

Early detection of individuals who are at high risk for developing type 2 diabetes is a powerful strategy to tackle the diabetes epidemic, as it allows for targeted prevention [1]. Type 2 diabetes affects one in eight adults in the United States (US) and is the nation's seventh leading cause of death. A look beyond the averages reveals a disparity in the incidence and mortality rate of diabetes across racial groups, with non-Hispanic Whites being the only group wherein rates are below the national average [2, 3].

**Funding:** HTC and AK are supported by a grant from Novo Nordisk Foundation Challenge Programme for the project Harnessing the Power of Big Data to Address the Societal Challenge of Aging (NNF17OC0027812). TOA is supported by a grant from the Independent Research Fund Denmark (7025-00005B). TVV is supported by the "Data Science Investigator - Emerging 2022" grant from Novo Nordisk Foundation (NNF22OC0075284). The funders had no role in study design, data collection and analysis, decision to publish, or preparation of the manuscript.

**Competing interests:** The authors have declared that no competing interests exist.

Despite their comparatively lower risk [4], non-Hispanic White groups remain overrepresented in the diabetes risk prediction literature [5]. Consequently, the implementation of evidence-based risk prediction in primary care is vulnerable to limited generalizability to other racial groups. Biased prediction models may prioritize individuals of certain racial groups for preventive action at different rates, or at different stages in their disease progression [6]. Such unequal predictions would exacerbate the systemic healthcare inequalities we are currently seeing, which stem from socioeconomic inequalities, differential health literacy and access to healthcare, and various forms of discrimination between majority and minority populations [7]. For instance, biased type 2 diabetes prediction models that fail to identify at-risk individuals among racial minorities accurately would likely sustain or worsen the current disparities in developing diabetic complications [8].

Today, the current standards of care focus on identifying individuals with prevalent type 2 diabetes or prediabetes in asymptomatic adults and referring them to appropriate prevention or treatment routes [10]. Screening in the US relies mainly on the Prediabetes Risk Test (PRT) [10, 11], a diagnostic model developed by the National Diabetes Prevention Program (https://diabetes.org/diabetes/risk-test and https://www.cdc.gov/prediabetes/risktest). To the best of our knowledge, despite the availability of numerous US-based *prognostic* risk prediction models [7–9] and the success of prognostic models forecasting cardiovascular diseases [9], very few or none of the prognostic type 2 diabetes models are currently implemented in routine clinical practice. Prognostic models for type 2 diabetes, incorporating information on demographic, socioeconomic, lifestyle, familial, and biological factors, can be used to calculate a *personalized* probability of developing the disease in a certain time window [10]. The United Kingdom-based '*Healthier You*: *National Health Services Diabetes Prevention Programme*', for example, has allowed the identification of thousands of individuals at high risk of developing type 2 diabetes, thereby enabling early intervention that successfully reduced their type 2 diabetes risk via structured support, adoption of healthier lifestyles, and reductions in adiposity and glycemia [11]. Consequently, we do expect the implementation of prognostic T2D prediction models in the US, as well as other countries, in the near future, largely due to financial incentives; the prevention of T2D and its comorbidities (e.g., obesity and cardiovascular diseases) and complications will be more cost-effective in the long run than the treatment of them [12]. Furthermore, the wide implementation and success of prognostic predictive models for cardiovascular diseases have already paved the way for how such models can be widely utilized to prevent disease [9].

In this project, we set out to investigate whether the PRT [13] and two key US-based prognostic prediction models for type 2 diabetes, demonstrate consistent (mis)calibration across Black and White non-Hispanic US residents in the Continuous National Health and Nutrition Examination Survey (NHANES) survey [14]. We further investigate to what extent the current clinical guidelines may address or perpetuate these biases and discuss concrete action points for stakeholders to (further) promote fair prognostic disease prediction.

## Research design and methods

### Study population

The NHANES is a repeated cross-sectional survey that has been providing a vast amount of publicly available health data on a nationally representative sample of the civilian, non-institutionalized US population since 1999 [14]. Demographic, questionnaire, physical examination, and biochemical data are collected in two-year intervals. The reliability of sub-group data is ensured by the oversampling of minority racial groups and persons over the age of 60.

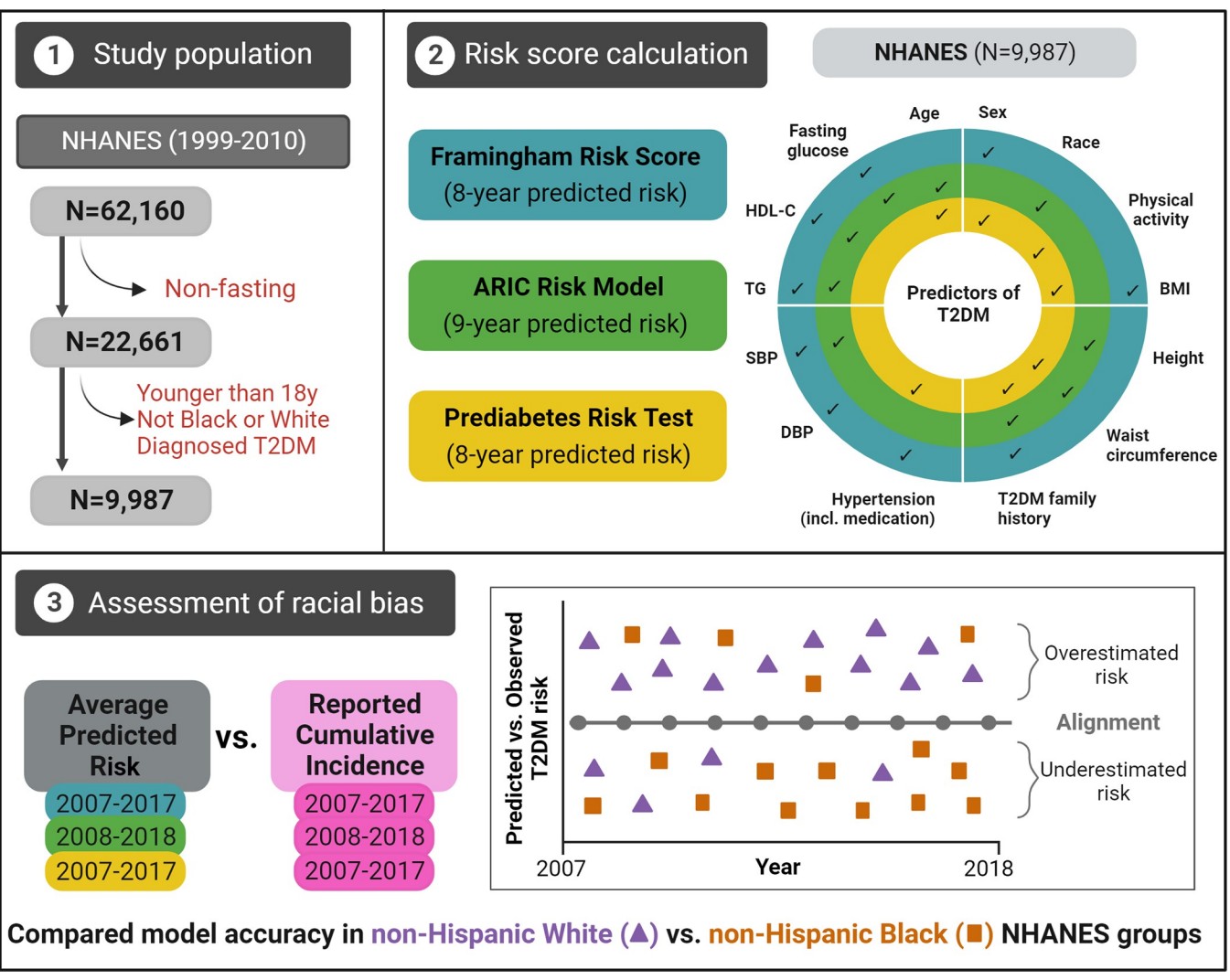

**Fig 1. Study design.**

Our population of interest comprised non-Hispanic White and non-Hispanic Black adults at risk of type 2 diabetes. We included NHANES survey content between 1999 and 2010, sampled in six independent 2-year batches (1999–2000, 2001–2002, . . ., 2009–2010). A total of 62,160 participants were surveyed during this timeframe, of whom 22,661 provided fasting blood samples. Analyses were restricted to the latter group due to the utilization of fasting biochemical measures in the prediction algorithms tested. Upon excluding individuals with diagnosed type 2 diabetes (based on self-report and medication use), those younger than 18 years, and those who did not self-report their race as non-Hispanic White or non-Hispanic Black (n = 12,674), we retained an analytical sample of 9,987 individuals (**Fig 1, S1 Fig**). We used fasting subsample weights to correct for representativeness in our analyses and allow for inferences made from this subset to be representative of the larger US population of interest.

## Prediction models

Our analyses focused on the PRT [13] issued by the National Diabetes Prevention Program, and two widely known North-American prognostic type 2 diabetes risk models: the

Framingham Offspring Risk Score [15] and the Atherosclerosis Risk in Communities (ARIC) Risk Model [16] (**S1 Table**). We used the NHANES database to extract the data needed to calculate model estimates according to the authors' guidelines (**Fig 1**). Continuous predictors included age (years), body mass index (BMI, kg/m$^2$), waist circumference (cm), height (cm), systolic and diastolic blood pressure (mmHg), fasting glucose (mmol/L), high-density lipoprotein cholesterol (HDL-C) (mmol/L), and triglycerides (mmol/L). Binary predictors were sex (male/female), race (non-Hispanic Black and non-Hispanic White), anti-hypertensive medication use (yes/no), previously diagnosed hypertension (yes/no), physical activity (yes/no, coded yes for datasets between 1999–2006 if the participant responded "More active" to a single question "Compared with most men/women your age, would you say that you are . . .", and coded yes for datasets between 2007–2010 if the participant reported either occupational or leisure-time vigorous physical activity), and family history of diabetes (yes/no; coded yes for datasets between 1999–2004 if participants answered in an affirmative to the presence of diabetes in any of the specific family members, and coded yes for datasets between 2005–2010 if participants answered yes to a single question inquiring whether a close relative has diabetes). For the Framingham Offspring Risk Score and ARIC Risk Model, we included the latter-described variable as a proxy for *parental history of diabetes* due to the unavailability of a better-defined variable in NHANES. The three investigated models included the following predictors:

1. PRT: age, sex, BMI, waist circumference, family history of diabetes, history of hypertension, physical activity.

2. Framingham Offspring Risk Score: sex, fasting glucose, BMI, HDL-C, family history of diabetes, triglycerides, systolic and diastolic blood pressure, anti-hypertensive medication use.

3. ARIC Risk Model: age, race, family history of diabetes, fasting glucose, systolic blood pressure, waist circumference, height, HDL-C, triglycerides.

## Statistical analysis

All analyses were undertaken using R 4.1.2 [17]. The NHANES data were extracted and prepared for analysis using the framework depicted in **S1 Fig**. Missing data were imputed using multivariate imputation by chained equations [18]. For all variables, random forest was utilized (five iterations), and 15 imputed copies were generated. The imputation models included the sampling weights to account for the complex survey design [19]. Convergence was visually inspected for randomly selected imputed datasets. For race-specific analysis, the cohort was stratified to non-Hispanic Whites (n = 8,674), and non-Hispanic Blacks (n = 1,313). Race- and survey-specific estimates derived from the final analytical cohort (N = 9,987) from the multiple imputation framework were pooled using Rubin's rules [20].

We used the Framingham Offspring Risk Score and the ARIC Risk Model to predict the risk of type 2 diabetes for the study participants. Given those predictions, we subsequently calculated race- and year-specific average predicted incidence proportions and 95% confidence intervals (CIs) for eight or nine years (depending on the model used) after each survey batch interval. Following the statistical guidelines of NHANES, these race- and year-specific averages were weighted to account for the survey design of the NHANES data [21]. We also determined the proportion of each survey batch that would be identified for further screening based on their PRT score. The PRT score assigns a score between -1 and 9, with values of 5 or higher indicating a high risk of either prediabetes or type 2 diabetes. Cumulative type 2 diabetes incidences (matched to the respective risk score time frames) by race were calculated as the sum of yearly reported incidences extracted from the US Diabetes Surveillance System database [22].

We subsequently compared the latter with the NHANES-derived weighted average predicted risks of the investigated models. As an example, we used the overall and race-stratified NHANES population data from 1999–2000 to calculate 8-year average predicted incidence proportions using the Framingham Offspring Risk Score. We then compared these proportions to the estimates extracted from the US Diabetes Surveillance System database eight years after the study period (in this case, 2007).

For illustration purposes, we also plotted the observed cumulative incidences against the predicted proportion of undiagnosed type 2 diabetes cases of the PRT test (**Fig 1**)**.** Thus, we evaluated the predictive performance of the model in terms of overall calibration within different racial subgroups, by calculating expected-to-observed incidence risk ratios [23].

### Ethics statement

The NHANES was subject to ethics review by the National Center for Health Statistics Research Ethics Review Board and approval was obtained bi-annually between each full proposal review. All participants provided informed consent.

### Role of the funding source

This project was supported by the Novo Nordisk Foundation and the Independent Research Fund Denmark. The funders had no role in the study design; in the collection, analysis, and interpretation of data; in the writing of the report; and in the decision to submit the paper for publication.

## Results

Race-stratified weighted, unimputed descriptive statistics of the study population are presented per survey year in **Table 1**. Imputed descriptive statistics are shown in **S2 Table**. Compared to non-Hispanic White NHANES participants, non-Hispanic Black groups were younger, less likely to have resided in the US from birth or to have obtained a high school diploma, and more likely to be hypertensive or have a family history of diabetes. While non-Hispanic Blacks were also more likely to have a higher BMI than their White counterparts, they had more favorable triglyceride profiles and did not differ from non-Hispanic White groups in terms of waist circumference or HDL-C. For survey years spanning 1999–2004, non-Hispanic Whites were proportionally less physically active than non-Hispanic Blacks, with the trend reversing from 2005 onward.

**S3 Table** presents the race-stratified age-adjusted type 2 diabetes incidence rates for non-Hispanic Whites and non-Hispanic Blacks from the US Diabetes Surveillance System [22]. Overall, type 2 diabetes incidence peaked in 2008 and has been decreasing since, although incidences remain higher than what they were in 2000. There continues to be considerable variation across racial groups, with non-Hispanic Blacks consistently having higher incidence rates compared to non-Hispanic Whites.

Predicted average risks for each examined type 2 diabetes model were compared with cumulative incidences calculated from the US Diabetes Surveillance System. As an example, we used the NHANES cohort from 1999 to calculate race-stratified 8-year average predicted type 2 diabetes risk, and thus, calculated that by 2007, 7% of non-Hispanic Whites, and 6% of non-Hispanic Blacks are expected to develop type 2 diabetes. We compared our calculated estimates with real-life race-stratified cumulative incidences from the matching 8-year period (2000–2007) (cumulative incidences for this period are 5% for non-Hispanic Whites and 8% for non-Hispanic Blacks). We repeated this analysis for the three models, and each of the six

**Table 1. Descriptive statistics of the unimputed NHANES data (N = 9,987).**

| | 1999–2000 | | 2001–2002 | | 2003–2004 | | 2005–2006 | | 2007–2008 | | 2009–2010 | |
|---|---|---|---|---|---|---|---|---|---|---|---|---|
| | non-Hispanic White | non-Hispanic Black | non-Hispanic White | non-Hispanic Black | non-Hispanic White | non-Hispanic Black | non-Hispanic White | non-Hispanic Black | non-Hispanic White | non-Hispanic Black | non-Hispanic White | non-Hispanic Black |
| **N** | 1,222 | 175 | 1,575 | 231 | 1,451 | 221 | 1,450 | 221 | 1,474 | 224 | 1,502 | 241 |
| **Age**, years | 44.9 (17.1) | 40.2 (14.5) | 45.4 (16.9) | 39.9 (15.5) | 45.6 (17.2) | 40.1 (16.8) | 46.4 (17.5) | 40.7 (16.2) | 46.5 (17.3) | 40.8 (15.6) | 47.1 (17.5) | 41.1 (15.5) |
| **Female**, % | 52.2 | 54.2 | 52.0 | 55.0 | 52.1 | 55.6 | 50.9 | 54.8 | 51.7 | 54.4 | 52.0 | 54.3 |
| **High school graduate**, % | 82.8 | 62.1 | 84 | 69.2 | 84.6 | 66.1 | 86.5 | 71.7 | 83.9 | 70.6 | 85.8 | 71.9 |
| Missing, % | 3.5 | 4.9 | 3.5 | 5.1 | 4.5 | 6.1 | 3.4 | 7.5 | 3.5 | 6.1 | 3.0 | 4.5 |
| **Born in the US**, % | 95.0 | 86.6 | 95.8 | 92.4 | 94.4 | 93.9 | 95.5 | 90.8 | 96.1 | 92.0 | 94.0 | 84.5 |
| **Physically active**, % | 37.5 | 40.7 | 35.7 | 37.9 | 34.0 | 38.9 | 38.2 | 37.1 | 43.2 | 42.5 | 40.1 | 37.2 |
| **Family history of diabetes**, % | 44.5 | 46.4 | 46.5 | 46.9 | 45.5 | 52.4 | 34.7 | 41.9 | 31.4 | 37.9 | 29.8 | 42.7 |
| Missing, % | 5.3 | 5.2 | 5.3 | 6.7 | 6.7 | 7.5 | 6.5 | 8.6 | 5.6 | 6.9 | 4.8 | 5.3 |
| **Body mass index**, kg/m² | 26.9 (5.8) | 29.1 (7.1) | 27.7 (6.1) | 28.8 (7.6) | 27.8 (6.1) | 29.9 (7.6) | 28.2 (7.0) | 30.3 (7.8) | 27.6 (5.8) | 29.1 (7.1) | 28.2 (6.4) | 30.5 (8.1) |
| Missing, % | 0.5 | 1.8 | 6.6 | 3.6 | 1.4 | 2.0 | 1.7 | 2.2 | 1.9 | 2.1 | 1.2 | 0.4 |
| **Waist circumference**, cm | 93.5 (15.4) | 94.9 (16.6) | 95.5 (15.4) | 93.9 (16.7) | 96.9 (15.6) | 97.1 (17.0) | 97.0 (16.6) | 98.0 (17.5) | 96.3 (15.1) | 96.2 (16.2) | 97.8 (16.0) | 98.1 (16.9) |
| Missing, % | 1.7 | 4.1 | 5.2 | 4.2 | 4.6 | 6.7 | 4.5 | 4.9 | 6.6 | 5.4 | 4.4 | 5.2 |
| **Height**, cm | 170.3 (9.9) | 169.9 (9.5) | 170.1 (9.9) | 169.6 (9.9) | 170.8 (9.9) | 169.7 (9.4) | 170.4 (9.9) | 169.6 (9.7) | 170.8 (10.1) | 170.5 (9.0) | 170.5 (9.7) | 169.0 (9.7) |
| Missing, % | 0.5 | 1.8 | 3.6 | 1.2 | 1.3 | 2.0 | 1.6 | 2.0 | 1.8 | 2.1 | 0.9 | 0.4 |
| **Fasting glucose**, mmol/L | 5.1 (1.0) | 5.2 (1.5) | 5.1 (0.9) | 5.1 (1.0) | 5.2 (0.7) | 5.2 (0.9) | 5.2 (1.0) | 5.1 (0.8) | 5.1 (0.7) | 5.1 (1.1) | 5.2 (0.8) | 5.2 (0.8) |
| Missing, % | 5.6 | 10.4 | 4.4 | 13.5 | 5.5 | 6.1 | 5.3 | 9.4 | 4.1 | 14.7 | 5.1 | 9.7 |
| **HDL-C**, mmol/L | 1.3 (0.4) | 1.4 (0.4) | 1.3 (0.4) | 1.4 (0.4) | 1.4 (0.4) | 1.5 (0.4) | 1.5 (0.4) | 1.5 (0.5) | 1.4 (0.4) | 1.6 (0.4) | 1.4 (0.4) | 1.4 (0.5) |
| Missing, % | 5.7 | 10.8 | 4.5 | 14.1 | 5.2 | 5.9 | 5.1 | 8.8 | 4.2 | 14.1 | 4.9 | 9.5 |
| **Triglycerides**, mmol/L | 1.6 (1.1) | 1.1 (0.7) | 1.7 (2.1) | 1.2 (0.7) | 1.7 (1.4) | 1.2 (0.8) | 1.6 (1.2) | 1.2 (1.2) | 1.5 (1.1) | 1.0 (0.6) | 1.4 (1.1) | 1.1 (0.7) |
| **Triglycerides**, ln (mmol/L) | 0.3 (0.5) | 0.0 (0.5) | 0.3 (0.6) | 0.0 (0.5) | 0.3 (0.6) | 0.0 (0.5) | 0.3 (0.6) | 0.0 (0.5) | 0.2 (0.5) | -0.1 (0.6) | 0.2 (0.5) | 0.0 (0.5) |
| Missing, % | 5.9 | 11.0 | 4.5 | 14.1 | 5.2 | 6.3 | 5.3 | 9.9 | 4.2 | 14.1 | 4.9 | 9.5 |
| **Systolic BP**, mmHg | 121.2 (17.7) | 123.2 (17.8) | 122.4 (18.8) | 124.1 (19.4) | 121.2 (17.9) | 124.8 (20.7) | 121.2 (17.2) | 125.3 (18.0) | 119.2 (16.0) | 121.5 (17.3) | 118.2 (15.5) | 122.9 (16.7) |
| **Diastolic BP**, mmHg | 72.0 (11.9) | 74.1 (12.2) | 72.6 (11.5) | 73.4 (13.2) | 70.4 (13.1) | 71.2 (12.8) | 68.8 (12.4) | 69.8 (14.7) | 69.3 (11.6) | 70.4 (12.5) | 67.8 (11.7) | 71.1 (13.5) |
| Missing, % | 2.4 | 6.5 | 3.5 | 5.9 | 5.1 | 5.5 | 3.8 | 8.5 | 4.6 | 6.2 | 3.6 | 7.6 |
| **Antihypertensive use**, % | 14.9 | 15.4 | 16.1 | 19.1 | 18.4 | 18.4 | 18.5 | 21.8 | 21.2 | 22.4 | 21.3 | 21.4 |
| Missing, % | 27.1 | 28.6 | 28.2 | 28.6 | 24.8 | 24.8 | 22.2 | 21.5 | 18.2 | 17.3 | 17.5 | 22.1 |
| **Hypertension diagnosis**, % | 41.4 | 44.3 | 43.7 | 46.9 | 41.4 | 41.4 | 39.7 | 44.2 | 37.1 | 41.5 | 37.0 | 44.4 |

Data displayed as means and standard deviations for continuous variables and percentages for categorical variables.

Abbreviations: BP–blood pressure; HDL-C–high-density lipoprotein cholesterol.

NHANES fasting survey weights were used to estimate weighted means and variances.

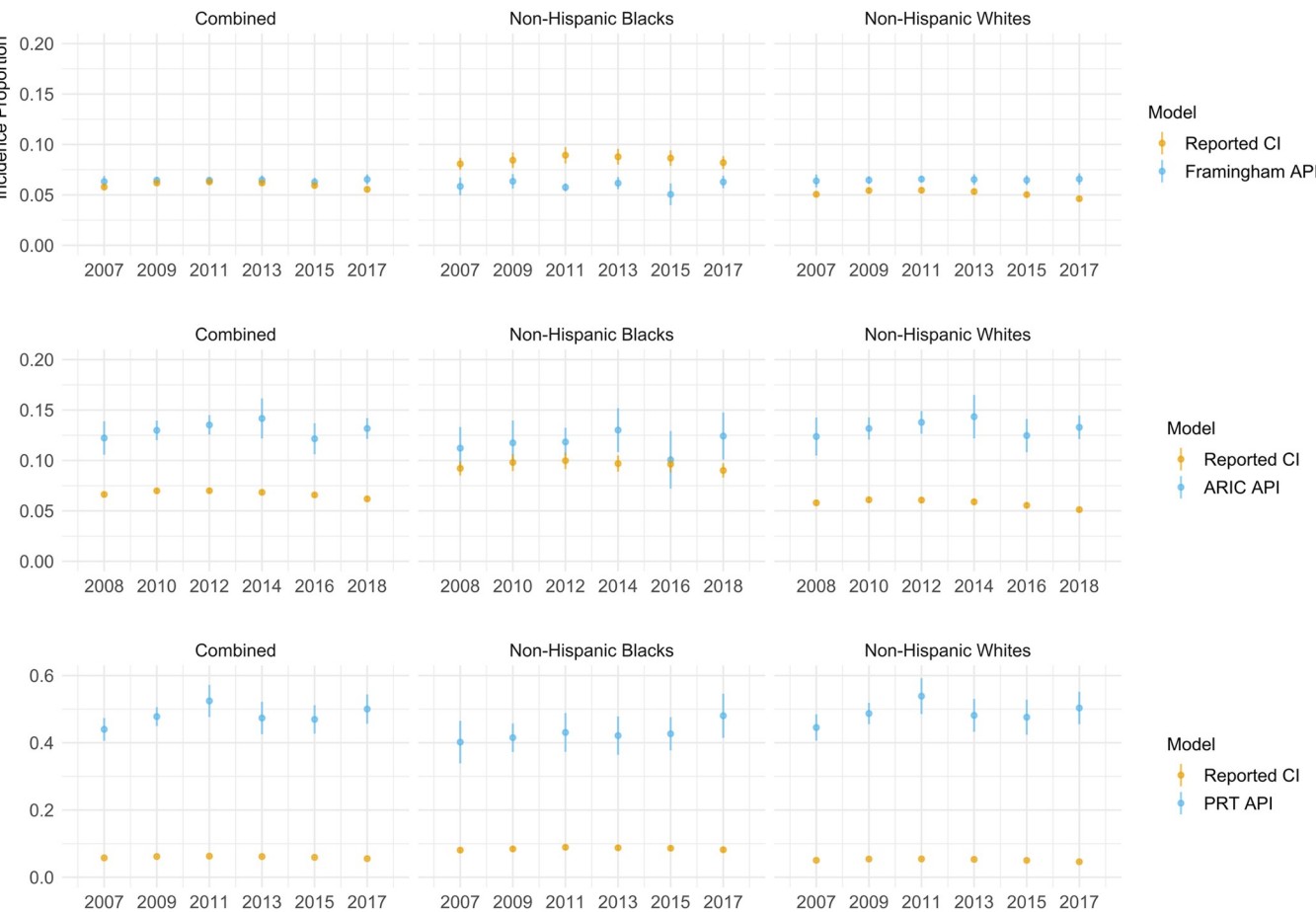

**Fig 2. The first and third panel show the 8-year predicted type 2 diabetes risk for the Framingham Offspring Risk Score and the Prediabetes Risk Test and 8-year cumulative incidences of type 2 diabetes from the US Diabetes Surveillance System, combined and per racial group.** The second panel shows the 9-year predicted type 2 diabetes risks for the ARIC Model and 9-year cumulative incidences of type 2 diabetes from the US Diabetes Surveillance System, combined and per racial group. Abbreviations: API—average predicted incidence (proportion); CI—cumulative incidence; PRT–Prediabetes Risk Test. Error bars reflect 95% confidence intervals.

cohorts from NHANES between 1999 and 2010, to obtain race-stratified predicted estimates until 2017 (**S4 Table**). Results for the three models are shown in **Fig 2**.

While the Framingham Offspring Risk Score overestimated type 2 diabetes risk for non-Hispanic Whites, it underestimated the risk for non-Hispanic Blacks. The PRT and the ARIC Model overestimated type 2 diabetes risk for both races, but more so for non-Hispanic Whites compared to non-Hispanic Blacks. The PRT showed the highest overestimation; based on the scores approximately half of the total populations of the cohorts were prioritized for screening.

We calculated the ratios of the calculated average predicted risks (incidence proportion) to the calculated cumulative incidences (**Fig 3**). All three models demonstrated an overestimation of risk for non-Hispanic Whites. The ARIC model delivered a lower overestimation for non-Hispanic Blacks (average ratio = 1.22) compared to non-Hispanic Whites (average ratio = 2.31). Similarly, the PRT delivered a lower overestimation for non-Hispanic Blacks (average ratio = 5.05) compared to non-Hispanic Whites (average ratio = 9.51) when considering the 8-year risk of diabetes.

We visualized the correspondence between the dichotomized PRT scores of individuals and the individual predicted probabilities from the two prognostic type 2 diabetes risk

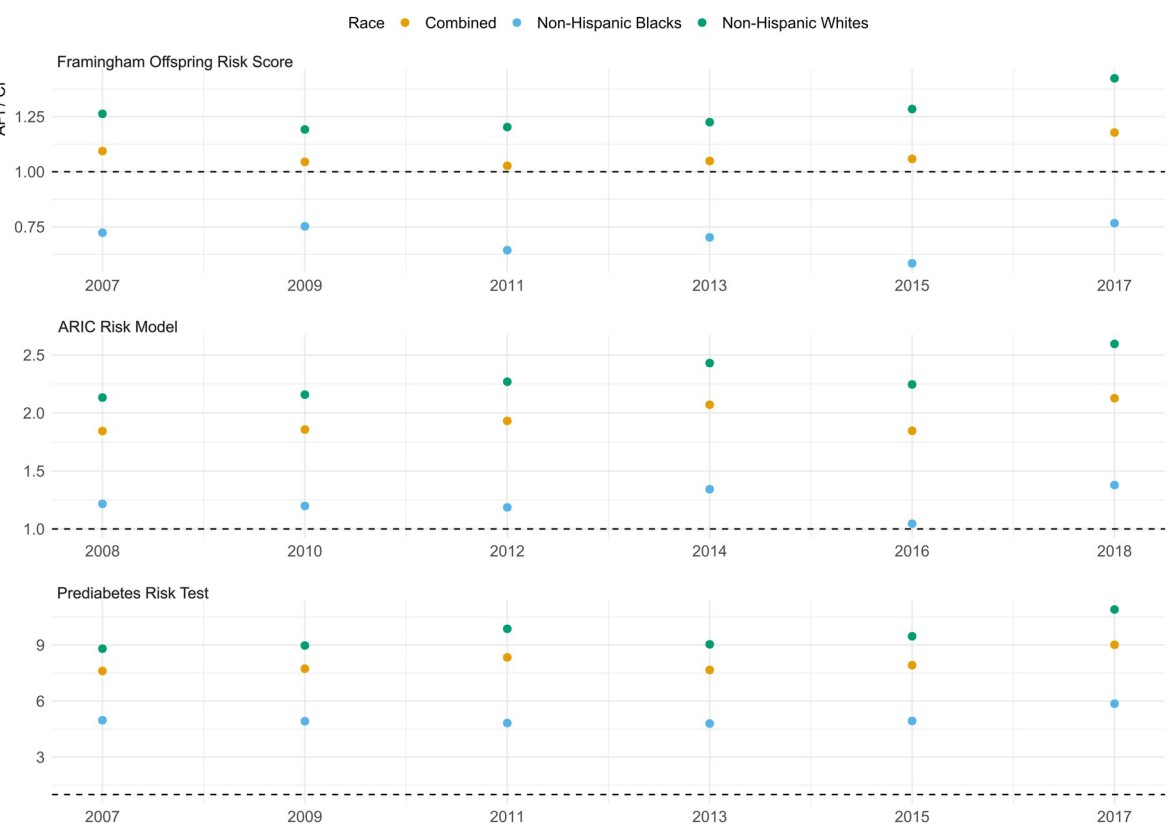

**Fig 3. Ratios between predicted Framingham Offspring Risk Score, ARIC model, and Prediabetes Risk Test incidence proportions and reported cumulative incidences of type 2 diabetes overall, and per racial group.** Abbreviations: API—average predicted incidence (proportion); CI—cumulative incidence.

prediction algorithms (**Fig 4**). The PRT scores showed a moderate correlation with the predicted probabilities by the Framingham (Kendall's tau = 0.39) and the ARIC models (Kendall's tau = 0.56). Above and below the threshold of scoring five on the PRT, non-Hispanic Whites demonstrated a slightly higher risk of type 2 diabetes compared to non-Hispanic Blacks using both the Framingham and the ARIC models, however, this difference did not reach statistical significance (95% CIs overlapping 0).

## Discussion

In recent decades, algorithmic decision making has become a routine part of healthcare. Simple risk scores are routinely used to evaluate an individual's risk of developing a disease for most cardiometabolic outcomes, including type 2 diabetes [24], and complex artificial intelligence-driven models are continuously developed to predict common complications of type 2 diabetes [25]. Nonetheless, regardless of how powerful artificial intelligence models can be in capturing complex interactions and patterns in data, the appropriateness of the available datasets in terms of representativeness and quality remains crucial to algorithmic design.

Recent examples have shown that models developed within biased datasets, or not directly addressing data biases, will likely propagate inequalities into clinical decisions [6, 26, 27]. As a result of biased algorithmic decision making, those who are already marginalized and less represented, may encounter further obstacles in accessing optimal healthcare, generating a vicious cycle. In this report, we investigated whether the PRT—a nationally adopted screening

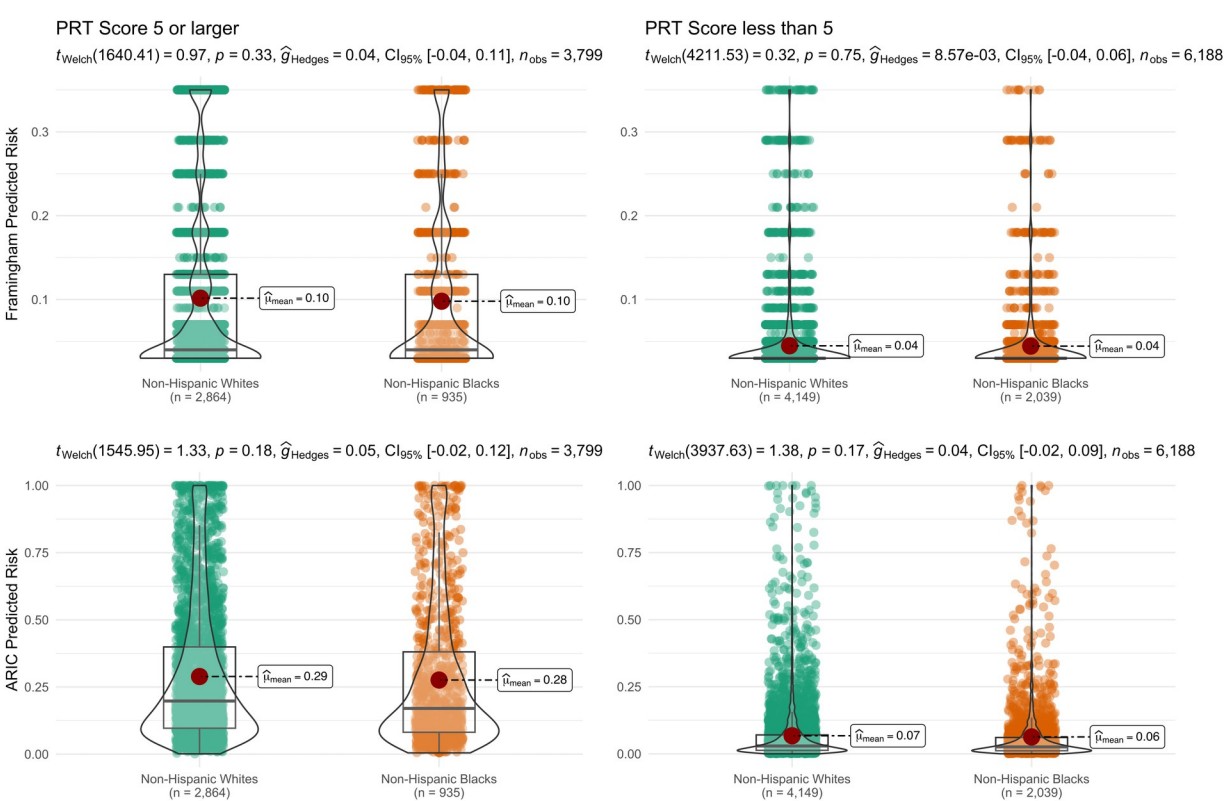

**Fig 4. Box-Violin Plots of Framingham Offspring Risk Score and ARIC Model individual predicted risks in groups scoring less or greater than 5 in the Prediabetes Risk Test.** Group differences were estimated using Welch's t-tests.

algorithm for prediabetes and type 2 diabetes—and two landmark type 2 diabetes predictive models developed in the US were racially biased. We compared the proportion of the population identified as being at high risk for incident type 2 diabetes with corresponding reported national statistics on type 2 diabetes incidence by racial group, within the predicted time horizon.

We interpret the model-derived proportions, indicating the population at risk for type 2 diabetes within a certain timeframe, as the fraction of the racial groups that could be prioritized for preventive intervention in that timeframe. When models underestimate the proportion of the population at risk for type 2 diabetes compared with national statistics, that indicates that the models identify fewer individuals in need of preventive action than the actual needs based on national statistics. Conversely, in case of overestimation, larger fractions of the population could be prioritized for preventive action and larger proportions would potentially receive support to achieve their health targets than would be necessary based on national statistics. Imbalances in over- and underestimation between racial groups reflect *algorithmic bias*, i.e., systematic differences in the performance of algorithms across specific groups. Overdiagnosis and overtreatment pose significant challenges, such as the detection and unnecessary treatment of individuals who will ultimately remain asymptomatic [28]. However, in the case of type 2 diabetes, the consensus on preventive action largely involves non-invasive lifestyle interventions, which are less likely to pose harm to false positives compared to e.g., pharmaceutical treatment. Nonetheless, imposing unnecessary, even nonaggressive, procedures on individuals, can lead to negative psychosocial consequences, e.g., increased stress levels, reduced quality of life, psychological and financial harms, and social labeling and stigma [29].

Ultimately, and given the nature and guidelines of the specific chronic disease, we consider a potential underestimation of risk as a bigger threat compared to an overestimation.

Our results show that the PRT and the two examined prognostic type 2 diabetes risk prediction models consistently demonstrated a larger risk overestimation for non-Hispanic Whites than Blacks. Moreover, the Framingham Offspring Risk Score, developed in a 99% White and non-Hispanic population, underestimated type 2 diabetes risk for non-Hispanic Blacks.

Although the nationally adopted PRT appears to be highly sensitive (i.e., very unlikely to misclassify diabetes cases as non-cases), it is also non-specific and identifies approximately half of the total population for the screening of type 2 diabetes, very likely falsely identifying individuals who are normoglycemic. Despite the wide net the PRT casts, this risk scoring still identifies proportionally more non-Hispanic Whites for preventive action than non-Hispanic Blacks. When compared to the observed national incidence rates, the PRT fails to fully account for the racial differences that we observe in real-life incidence rates. However, a certain amount of overestimation from PRT was expected, given that it also indicates cases of prediabetes.

Notably, all three examined algorithms almost exclusively account for inherent risk (biological sex and family history of diabetes) or markers of metabolic health (biochemical profiles, blood pressure, and anthropometry). The ARIC Risk Model was the only algorithm to include race, a non-metabolic parameter; and the PRT was the only algorithm incorporating a lifestyle factor (physical activity). In this nationally representative dataset, the three examined models consistently predicted higher average risks for non-Hispanic Whites compared to non-Hispanic Blacks, contrary to the official national statistics. This observation indicates that the metabolic health variables implemented in the models failed to capture the observed lower risk of type 2 diabetes in non-Hispanic Whites compared to Blacks. While our results are only suggestive, the observed phenomenon could be explained by the absence of socio-economic determinants in the models such as health literacy status and access to healthcare. As non-Hispanic Whites have higher socio-economic status on average, these factors might act as compensatory, protective factors that would result in lower realized risk of type 2 diabetes compared to non-Hispanic Blacks. Our descriptive statistics of the analyzed data confirm these patterns, with non-Hispanic Blacks having lower educational attainment on average and a higher likelihood of being immigrants. As an acute solution to the observed algorithmic bias resulting from these health inequalities, the investigated algorithms–and especially the PRT that is already adopted by healthcare–would likely benefit from the explicit inclusion of (additional) markers related to education, health literacy, and other socioeconomic determinants, that are expected to correlate with race [30, 31]. The inclusion of such features in the models would likely shift the distributions of predicted probabilities higher for non-Hispanic Blacks and thus prioritize a larger fraction of this population for preventive action.

Given the underrepresentation of minorities in studies where models were developed, limited algorithmic generalizability might be expected within some racial groups. When it comes to sample size considerations in developing novel algorithms, three major approaches are to be considered. The first approach is to develop models in nationally representative populations. This will result in predictive models that will perform more accurately for the majority. The cohort investigated by the developers of the ARIC Model [16] most closely represented this approach by comprising 85% non-Hispanic Whites and 15% non-Hispanic Blacks. In the original publication, the ARIC model did perform most accurately in the majority population, although the positive coefficient in the risk equation for Blacks increased the risk estimates for this group resulting in an accuracy nearing that of the majority population. When tested externally using NHANES data, however, we still see marked differences in model performance between non-Hispanic Whites and Blacks, reflecting algorithmic bias. The second approach is

to develop models in cohorts with roughly equal sample sizes across groups, resulting in models that are expected to perform with the same confidence and precision across groups, but may perform less optimally for the majority. The third approach is to develop separate models in separate groups. This approach was taken when developing the Reynolds Risk Scores separately for men [32] and women [33] to predict sex-specific 10-year cardiovascular disease risk. While both total sample size and sample size of respective groups are important considerations in the development of prediction models, societal and data biases (e.g., differential access to healthcare) also impact the development and subsequent performance of algorithms, as observed before [6].

Prediction models can perpetuate and reinforce data biases that lie in the core of the societies we live in. Explicit counteraction is warranted to address these biases when developing prediction models. In recent years, *algorithmic fairness*, as an emergent research area, has been receiving increasing attention, and the development of toolkits for the development of fair models has taken flight [34]. It has been raised that without the consideration of sensitive attributes, it is cumbersome to correct data biases and develop fair models [35]. However, the inclusion of race as a feature in predictive models has been at the center of academic debate [36–38]. For instance, the vast majority of clinical prediction models for cardiovascular diseases do not include race in the assessment of individual risk [36]. Arguments for the omission of these sensitive attributes are that racial discrimination can potentially be reinforced with the addition of race in the models (i.e., racial profiling) and that there is only weak evidence for genetic or biological differences across races [39]. Additionally, researchers have expressed concern that race insertion might create risks of falsely interpreting racial inequities as immutable facts rather than disparities that require intervention [40, 41]. Nonetheless, variables encoding race are deemed to reflect a complex combination of factors related to biological, cultural, as well as socio-contextual aspects. Therefore, while the use of race as a predictor may suffer from crucial flaws, it can also pave the way for better algorithmic decision-making across various racial groups, for example by promoting personalized care and targeting those in the most need of intervention [36]. The exclusion of race from algorithms may lead to differential external validity and potentially harmful decision making, as models may fail to account for differential risk across groups [39]. Undeniably, we are facing a 'data problem': data currently used to generate models are likely to embed inherent racial imbalances, lack intersectionality, and/or important social-contextual features that potentially confound the association between race and various clinical endpoints [38, 40]. As an example, Obermeyer *et al*. demonstrated significant racial bias in a predictive algorithm using healthcare expenditure as a predictive feature for prioritizing individuals for healthcare interventions [6]. Thus, it is crucial for future studies to investigate complex diseases through a 'structural racism lens' [42, 43] and disentangle causal relationships between various socioeconomic factors, human physiology, and outcomes, in pursuit of identifying the most appropriate candidate features for predictive modeling.

Based on these findings, we suggest that any published and/or candidate diagnostic or prognostic models should demonstrate algorithmic fairness before adoption in healthcare, e.g., via a systematic comparison of their performance in external samples stratified by sensitive attributes, such as race, ethnicity, and sex. Until fairness is a default consideration in algorithmic development, and models adopted by healthcare are thoroughly checked for fairness, societal inequalities will likely keep propagating into clinical decisions. Thus, we call for medical journals to recommend reporting criteria related to the fairness of novel predictive algorithms. While numerous toolkits, checklists, and governance frameworks have been developed on the topic [44], we emphasize five key points that journals may want to require reporting on: (i) potential data biases; (ii) possible measurements of bias; (iii) utilized bias mitigation strategies;

(iv) potential impacts of the biases; (v) and the generalizability of results across groups defined by relevant sensitive attributes. We recommend that expert groups and healthcare policy-makers responsible for the adoption of any risk models (e.g., cutoffs, simple risk scores, or complex algorithms) in clinical practice consider fairness a decisive factor. In the specific context of type 2 diabetes, there is a need to develop an accurate risk score for new-onset diabetes that helps identify groups with unmet needs within minorities.

Our study needs to be considered under some limitations. First, the predictive performance of the models could not be assessed in terms of discrimination as in their traditional reported form. The repeated cross-sectional design of the NHANES did not allow us to compare the individual predicted probabilities returned by the prediction models with the individual observed outcomes since participants were not followed-up over time. While prediction models are often used for decision-making at the individual level, one needs to bear in mind that predicted probabilities are drawn from groups of persons with comparable characteristics in terms of predictor values. To show disparities of predictive performance across racial groups, we reported ratios of average predicted to average observed type 2 diabetes incidences as an overall, summary measure of calibration within the different racial groups [23]. The interpretation of our findings thus remains at the group level. Second, we caution against an interpretation of the trend over time in the estimated predictive performance, as changes in observed incidence over time could be merely due to changes in case detection. Skewed case detection across racial groups (i.e., differential measurement error), could exaggerate or attenuate the reported disparities. This is, however, less of a concern due to the lack of observable differential detection in the available national diagnosed/undiagnosed type 2 diabetes data [45]. Third, the complex survey design of NHANES complicated the estimation procedure in the presence of missing item responses. To address this issue, we performed multiple imputation, while including the sampling weights in the imputation model [19], and using a random forest model to allow for complex interactions and non-linearities to be captured [46]. Last, we acknowledge that a large number of type 2 diabetes models are available, and it would be possible to extend our list of examined models. Here we selected three widely known models developed in the US, but future reports could include a wider range of models for testing.

In summary, our study shows that the PRT currently adopted by US healthcare, and prognostic type 2 diabetes prediction models available for adoption in US healthcare are likely attached with some degree of racial bias, which in turn is likely to perpetuate inequalities by providing fewer benefits to minorities, who already demonstrate higher risk for metabolic diseases. We provide specific recommendations that we believe can improve the standards of publishing and the adoption of algorithmic processes in healthcare.

## Supporting information

**S1 Fig. Study flowchart.**
(TIF)

**S1 Table. Type 2 diabetes risk prediction models included in the analytic framework.**
(XLSX)

**S2 Table. Descriptive statistics of the imputed NHANES data (N = 9,987).**
(XLSX)

**S3 Table. Age-adjusted incidence rates of type 2 diabetes per 1000 individuals.**
(XLSX)

**S4 Table. Predicted type 2 diabetes risks per racial group by the three prediction models, and cumulative incidences of type 2 diabetes from the US Diabetes Surveillance System.** (XLSX)

## Author Contributions

**Conceptualization:** Héléne T. Cronjé, Alexandros Katsiferis, Leonie K. Elsenburg, Thea O. Andersen, Naja H. Rod, Tri-Long Nguyen, Tibor V. Varga.

**Data curation:** Héléne T. Cronjé, Alexandros Katsiferis, Thea O. Andersen, Tibor V. Varga.

**Formal analysis:** Héléne T. Cronjé, Alexandros Katsiferis, Leonie K. Elsenburg, Thea O. Andersen, Tri-Long Nguyen, Tibor V. Varga.

**Funding acquisition:** Naja H. Rod, Tibor V. Varga.

**Investigation:** Héléne T. Cronjé, Alexandros Katsiferis, Leonie K. Elsenburg, Thea O. Andersen, Tri-Long Nguyen, Tibor V. Varga.

**Methodology:** Héléne T. Cronjé, Alexandros Katsiferis, Naja H. Rod, Tri-Long Nguyen, Tibor V. Varga.

**Project administration:** Tibor V. Varga.

**Resources:** Naja H. Rod.

**Supervision:** Naja H. Rod, Tibor V. Varga.

**Visualization:** Héléne T. Cronjé, Alexandros Katsiferis, Tibor V. Varga.

**Writing – original draft:** Héléne T. Cronjé, Alexandros Katsiferis, Leonie K. Elsenburg, Thea O. Andersen, Tri-Long Nguyen, Tibor V. Varga.

**Writing – review & editing:** Héléne T. Cronjé, Alexandros Katsiferis, Leonie K. Elsenburg, Thea O. Andersen, Naja H. Rod, Tri-Long Nguyen, Tibor V. Varga.

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
