## [Decision Letter · Decision Letter 0]

27 Mar 2023

PGPH-D-23-00034

Assessing racial bias in type 2 diabetes risk prediction algorithms

Dear Dr. Varga,

Thank you for submitting your manuscript to PLOS Global Public Health. After careful consideration, we feel that it has merit but does not fully meet PLOS Global Public Health’s publication criteria as it currently stands. Therefore, we invite you to submit a revised version of the manuscript that addresses the points raised during the review process.

We look forward to receiving your revised manuscript.

Kind regards,

Young-Rock Hong

Academic Editor

Journal Requirements:

1. We have noticed that you have uploaded Supporting Information files, but you have not included a list of legends. Please add a full list of legends for your Supporting Information files after the references list. 

Additional Editor Comments (if provided):

Reviewers' comments:

Reviewer's Responses to Questions

**Comments to the Author**

1. Does this manuscript meet PLOS Global Public Health’s publication criteria? Is the manuscript technically sound, and do the data support the conclusions? The manuscript must describe methodologically and ethically rigorous research with conclusions that are appropriately drawn based on the data presented.

Reviewer #1: Yes

Reviewer #2: Yes

2. Has the statistical analysis been performed appropriately and rigorously?

Reviewer #1: Yes

Reviewer #2: Yes

3. Have the authors made all data underlying the findings in their manuscript fully available (please refer to the Data Availability Statement at the start of the manuscript PDF file)?

Reviewer #1: Yes

Reviewer #2: Yes

4. Is the manuscript presented in an intelligible fashion and written in standard English?

Reviewer #1: Yes

Reviewer #2: Yes

5. Review Comments to the Author

Reviewer #1: Nice idea and topic very well presented by authors.

Here are my comments:

1- Introduction:

- More literature needs to be added to the intro explaining more details about the current prediction models, how they work, their impact and their relevance to NHANES data.

- More explanation about the impact of misdiagnosis of diabetes depending on race should be included showing recent references

2- Methodology:

- The process applied to NHANES data should be explained in more clarity mentioning the confounders and stratification of the sample in separate headings.

Reviewer #2: Overall, this study provides important insights into the potential racial bias in type 2 diabetes risk prediction models using NHANES data. The authors have followed a rigorous methodology and have presented their findings in a clear and accessible manner.

6. PLOS authors have the option to publish the peer review history of their article (what does this mean?). If published, this will include your full peer review and any attached files.

**Do you want your identity to be public for this peer review?** For information about this choice, including consent withdrawal, please see our Privacy Policy.

Reviewer #1: No

Reviewer #2: No

---

## [Editor Report · Decision Letter 1]

18 Apr 2023

Assessing racial bias in type 2 diabetes risk prediction algorithms

PGPH-D-23-00034R1

Dear Dr. Varga,

We are pleased to inform you that your manuscript 'Assessing racial bias in type 2 diabetes risk prediction algorithms' has been provisionally accepted for publication in PLOS Global Public Health.

Best regards,

Young-Rock Hong, PhD, MPH

Academic Editor